# Temperature Illusions in Mixed Reality using Color and Dynamic Graphics

Connor Wilson[*]
University of New Brunswick

Daniel J. Rea[†]
University of New Brunswick

Scott Bateman[‡]
University of New Brunswick

## ABSTRACT

Sensory illusions – where a sensory stimulus causes people to perceive effects that are altered by a different sensory stimulus – have the potential to enrich mixed-reality based interactions. The well-known color-temperature illusion is a sensory illusion that causes people to, somewhat counter intuitively, perceive blue objects to feel warmer and red objects to feel colder. There is currently little information about whether this illusion can be recreated in mixed reality (MR). Additionally, it is unknown whether dynamic graphical effects made possible by mixed-reality systems could create a similar or potentially stronger effect to the color-temperature illusion. The results of our study (n=30) support that the color-temperature illusion can be recreated in MR, and that dynamic graphics can create a new temperature-sensory illusion. Our dynamic-graphics-temperature illusion creates a stronger effect than the color-temperature illusion and has more intuitive relationship between the stimulus and the effect: cold graphical effects (a virtual ice ball) are perceived as colder and hot graphical effects (a virtual fire ball) as hotter. Our results demonstrate that mixed reality has the potential to create novel and stronger temperature-based illusions and encourage further investigation into graphical effects to shape user perception.

**Index Terms:** Human-centered computing—Human computer interaction (HCI)—Empirical studies in HCI; Human-centered computing—Human computer interaction (HCI)—Mixed / augmented reality—Mixed / augmented reality Computing methodologies—Computer graphics—Graphics systems and interfaces—Perception

## 1 INTRODUCTION

Sensory illusions – where a sensory stimulus causes people to perceive effects that are strengthened, contradicted, or overridden by a different sensory stimulus – have the potential to enrich mixed-reality based interactions (e.g., [2, 21, 24]). Illusions could be especially useful to help improve immersion or create novel experiences in mixed reality (MR) systems, which typically lack the ability to affect physical senses such as touch, temperature, or smell. There is little information about which known illusions can occur in MR environments and what new illusions might be possible. It could be that artistic or technical choices and limitations in MR systems may provide stimuli that lack a sufficient level of fidelity and realism and, as a result, may fail to alter our perceptions in the same way that real, physical stimuli can. Creating sensory illusions in mixed-reality environments (e.g., with virtual graphics interacting with the real world), however, could enrich experiences without employing, for example, specialized hardware.

One relatively well-known sensory illusion is the *color-temperature illusion* [12, 13]. While the color-temperature illusion

---

[*]e-mail: connor.wilson@unb.ca

[†]e-mail: daniel.rea@unb.ca

[‡]e-mail: scottb@unb.ca

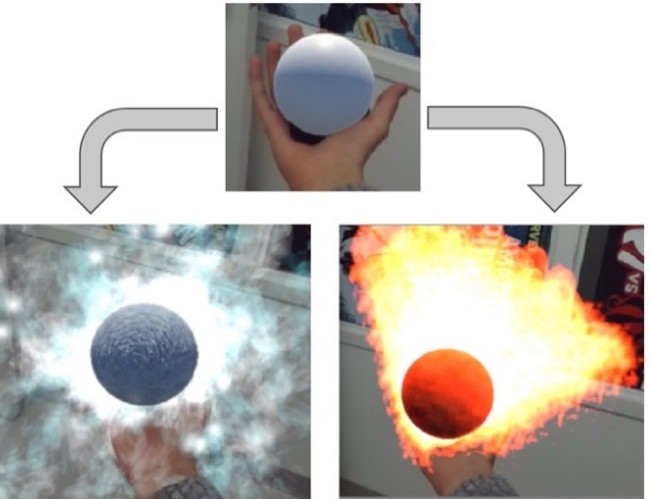

Figure 1: Top - a virtual ball shown using an augmented reality headset, used as a baseline condition in our study. Lower left - An icy particle and texture effect, which creates the illusion of the object feeling colder. Lower right - A fiery particle and texture effect, which creates the illusion of the object feeling hotter.

has created a reliable, modest effect on temperature judgments in controlled lab studies [12, 13, 31], it is yet unknown if this effect can be replicated with virtual objects using current MR technology. Further, it could be that MR technology could be used to create stronger or new temperature illusions due to its ability to create visual effects that are difficult to create in real life through graphical dynamic *textures* or *particle effects* (e.g., flames that can be held, but that do not change the actual temperature). Particle effects and static or dynamic textures are techniques from computer graphics that are commonly combined to visually recreate naturally occurring phenomena (for example, flames can flicker realistically on a glowing log) [33]. In this work, we explore the sensory effects of these graphical effects in MR.

In this work, we investigate if we can recreate the color-temperature illusion in mixed reality. We further investigate if we can strengthen the illusion with dynamic graphic effects using textures and particles. Specifically, we use augmented reality to place a virtual ball in people's hands that is either blue or red (color-temperature illusion) or a textured ball that is surrounded by frosty ice or flickering flames (using particle effects, see Figure 1). In all cases, the participant holds a real physical object with a known temperature, with a virtual ball (of a particular color or dynamic graphics effect) placed over top. Participants are then asked to estimate the object's temperature. If an illusion is present, it is expected that the estimates will be different from the real temperatures.

Through a study with 30 participants, we find that we can indeed replicate the color-temperature illusion using color in mixed reality. We also find that particle and texture-based graphics effects used together (which we refer to as *dynamic-graphics* for simplicity)

can also produce a change in perceived temperature. Interestingly the dynamic-graphics-temperature illusion creates an effect that is in the opposite direction of the color-temperature illusion. Thus, the mechanisms of the color-temperature illusion may differ from that of the dynamic-graphics-temperature illusion. While the color-temperature illusion leads people to perceive cooler colored objects are warmer even in mixed reality, the dynamic-graphics-temperature illusion makes people believe 'fiery' illusions are warmer than 'icy' ones. It further creates a stronger effect than the color-temperature illusion effect we observed. Our results also suggest that there may be limits for how long temperature illusions might last.

While texture and particle effects, as we characterize them, are common in many mixed reality experiences, we provide initial evidence that dynamic-graphics illusions can influence sensory perceptions in augmented reality. Our work contributes valuable new findings for the understanding of illusions, and we provide important new directions for the study and application of sensory illusions in mixed reality.

## 2 RELATED WORK

### 2.1 Visual Perception in Haptic Illusions

Haptic perception arises as the combination of tactile and proprioceptive senses and is strongly influenced by vision and hearing [29]. Because the range of systems involved, haptics perception is particularly susceptible to being influenced by illusions. While many definitions exist, [9], one view, which we adopt, is that an illusion is simply a perception that does not match reality [11].

The sense of sight is generally the most dominant sense and can override other senses [1] such as touch; for example, small bumps on an object's surface can be more easily perceived when a close-up image of the bumpy surface is shown [28]. An example of an illusion like where a visual stimuli leads to a haptic illusion is the size-weight illusion [26]. In this illusion, people are asked to compare the perceived weight of two objects of different size but same weight. The illusion will cause people to believe that the smaller object is heavier [26]. A common explanation of this illusion is the underlying expectation that the larger object will be heavier, creating the feeling that the larger object is lighter when lifted. Here, a person's expectations are primed by some property, and this expectation is broken by their experience, creating the illusion as the brain tries to resolve the stimuli. Other weight illusions have been identified and caused by objects' color, and perceived density [10, 16]. However, this is not universal as touch can sometimes lead to higher perceptual confidence than sight [8], and these relationships between perception and the senses is studied extensively in psychophysics (e.g., [17, 30, 32]).

It is possible to override illusions through learning, e.g., through exposure to different objects of various sizes and weights [4]. Even though learning reduces the effect of illusions, it has been shown that in the absence of other stimulus an illusion can persist (i.e., lifting the same object repeatedly will not erase the effect of an illusion) [7]. Further, when people have already learned a physical property of an object, it is more difficult for them to be affected by illusions [4].

### 2.2 Color-Temperature Illusion

The color-temperature illusion arises when people compare the temperature of objects of different colors. When comparing a "warm" colored object and a "cool" colored object of the same temperature, people tend to believe that the warm-colored object is colder than the cool-colored object [12]. It is believed that this illusion occurs due to people associating warmer colors with heat, so the expectation is for the object to be warm, leading people to believe the temperature is relatively cooler than their expectations (similar to the size-weight illusion). Previous research [12, 13] suggests that the warmer or colder the person believes the object will be before they pick it up, the greater the inverse temperature experienced will

be when held. While the color-temperature illusion has been widely reported, the exact strength of the effect seems low. Studies have reported difference in temperature perception to be on the order of 0.5°C [12], and have been studied in experiments where forced choice and relative temperature judgments are made (e.g., "is this object colder or warmer than the last?") [12, 25], making it difficult to understand the actual size of the illusion's effect) [6].

Warm colors include red, orange, and yellow, while cooler colors include blue and green [23]. For any of these illusions to have an effect, the viewer must have at least some abilities to see and distinguish between colors [25].

### 2.3 Applications of MR Induced Illusions

Sensory illusions studied in augmented and virtual reality are largely attributable to the dominance of sight over other senses. For example, in VR, subtle warping of movements can make users believe that physical object are placed somewhere else (called *haptic retargetting*) [1]. Similarly, people can experience body dysmorphia or other changes of self by altering aspects of an individual's virtual body and its movements [18, 27]. With AR advancements, illusion research can be more visually intricate and realistic, and can allow for visual manipulations of the body, objects, and environment to be performed [19, 20]. However, the expanded possibilities of AR systems and their ability to provide dynamic graphics been contrasted when recreating well-known illusions. Temperature illusions can occur based on other senses, such as sense of smell. Previous work has shown that stimulating the trigeminal nerve with scents can cause users to feel warmer/colder in virtual environments [3].

## 3 EXPERIMENT: TEMPERATURE ILLUSIONS

We were initially interested if we could recreate the color-temperature illusion using an augmented reality headset. We built an experimental system that allowed us to precisely control the temperature of a physical stimulus and then overlay it with a virtual white, red or blue ball, which should (according to the color-temperature illusion) lead to an illusion of a colder (for the red ball) or warmer (for the blue ball) temperature.

However, we were primarily interested in whether strengthening people's expectations (the theorized mechanism that leads to the color-temperature illusion) would lead to a stronger temperature illusion than the previously studied temperature illusion. Specifically, we investigated if holding a virtual flaming fire ball or virtual frosty ice ball (Figure 1) leads to a larger temperature illusion than experienced with color stimuli. We refer to these stimuli as dynamic graphics effects, and use a combination of textures and particle systems to create the flickering flames and condensed, frosty air.

Finally, we were also interested in how the illusions might degrade over time as people became accustomed to the stimulus provided. Therefore, our experimental design consisted of several blocks of trials that allowed us to observe any change in perceived temperature over time. We describe our experiment below, which was approved by the University of New Brunswick Research Ethics Board (on file as REB #2019-091).

### 3.1 Apparatus

We built a simple augmented reality system in Unity, using an HTC Vive Headset connected to a ZED Mini Camera that provided pass-through video AR. The system accurately placed balls of three different colors (white, red, and blue) and the two different particle effects (fire and ice) in a participant's hand. The particle effect conditions also had a base color (black for fire and white for ice), which where decided by early piloting that suggested it was the most realistic looking for the given effect. Further, the particle effects frequently overlapped the base sphere, making it appear more blue or orange (Figure 1).

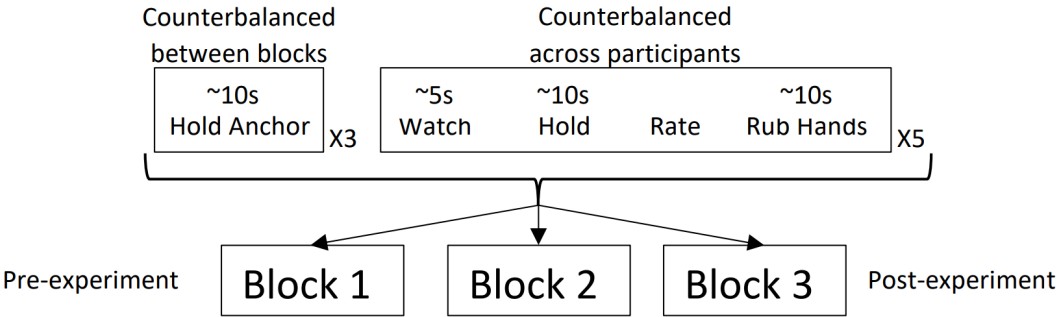

Figure 2: An overview of our multi-block experiment procedure. Each block contains the same structure: an anchoring phase with different temperature pads, and then an interaction with and a temperature estimation of each of the 5 dynamic graphical effects we were testing where the pads were secretly all the same temperature. There were three blocks total, one after the other, along with questionnaires at the beginning and end of the experiment. Brief breaks of a few minutes were allowed between blocks.

In order to accurately measure an illusion in temperature, we needed a method to control and calibrate the temperatures of objects. Twelve identical hot/cold packs were used to act as a temperature stimulus. Six of these packs were kept at room temperature (21°C), three were cooled to 7°C using a refrigerator, and three were warmed to 33°C using a sous vide machine in a pot of water (packs were quickly and thoroughly dried before use to remove any sensation of wetness), and pad temperatures were verified using an infrared thermometer before use. These temperatures were chosen as they were within the range of temperatures human beings can distinguish between [22]. We strictly controlled and measured the temperature so that we could be certain of the reference points given to the participant, and to know from what specific baseline temperature participants were experiencing when being influenced by our illusions, as we do not know if a variation in object temperature would affect our illusions.

In order to display the ball in 3D position accurately in MR, a Vive Tracker was placed on top of the hot/cold pack before handing it to the participant for the system to accurately place the virtual ball, which was displayed through the headset. Participants faced a white wall during the experiment.

### 3.2 Procedure

Upon arriving, participants completed a consent form, a demographics questionnaire and were explained the procedure summarized in Figure 2. Participants were informed that they would be given 15 different objects to hold and were asked to guess its temperature as accurately as possible. They were informed that the temperatures of all objects would be in the range of 7°C to 33°C. In reality, this was misleading since the actual 15 tests all used the same object temperatures.

The experiment consisted of 15 trials total divided evenly over 3 blocks allowing for 1 trial for each of the 5 different visual effects (red, blue, white, fire, and ice) per block (i.e., 5 trials per block, one for each color). The order of the effects for each participant was decided by counterbalancing using a Latin Square, with the participant's ordering being the same for all three blocks.

At the beginning of each 5-trial block, participants were given sample objects to hold for approximately 10 seconds while wearing the mixed reality headset, so they could familiarize themselves with the temperature range we described the experiment would use. There were three of these anchor packs, one whose real temperature was set to 7°C, one at 21°C, and one at 33°C. The order these were presented were counterbalanced between participants and blocks with a 3x3 Latin Square (temperature by block). After trying each of the three temperature packs, they would proceed through each of the 5 trials in the block.

During each trial, the experimenter would first physically show the participant the visual effect to the participant, by holding the temperature pack where they could easily view it, for 3-5 seconds to provide a brief exposure to the visual effect without physical sensation. Participants were then handed the object and they were asked to vocally estimate the temperature of the virtual ball that they were holding as accurately as possible, which the experimenter recorded. Participants were able to feel the pad for as long as they wished, but were asked after 10 seconds if they were ready to move onto the next trial. No participant took noticeably longer than this to estimate the temperature of any one object.

Importantly, *all 15 packs for the experimental trials were 21°C* (the 5 packs given out to test the 5 effects in each of the 3 blocks). In other words, only the anchor packs had different temperatures, and participants were led to believe there would be temperature variation in the 5 trials per block (since they were told that all backs would be between the cool anchor temperature of 7°C and the warm anchor temperature of 33°C), but there was not actually a difference. This was done so that all participant judgements were made from a controlled baseline temperature, and that the temperature was roughly in the middle of the range demonstrated by the anchor pads. Further, having room temperature objects is the a realistic scenario where a temperature illusion would most likely used – to make something that has no inherent ability to heat or cool itself feel hotter or colder. For example, holding a game controller that is at room temperature.

After their guess, the participant would return the object and were asked to lightly rub their hands together for 10 seconds to ground the sensation in their hands, rather than any perceived temperature from the object they were just holding. The experimenter then continued with the next trial using the next color or graphical effect stimulus.

After each five-trial block, the camera pass-through program that enabled augmented reality on our virtual reality headset was turned off so that the participant was in a fully virtual environment, unable to see the real environment around them. While this occurred, the experimenter walked to a different room to appear as though they were exchanging all the pads and came back with the anchor pads again to begin the next block (starting again with the anchoring phase). Participants were allowed a brief, couple minute break after each block. Once the participant completed all 15 experimental trials, they were asked to fill out an exit questionnaire and given their honorarium. Each experiment took approximately 30 minutes. The experiment procedure is summarized in Figure 2.

### 3.2.1 Motivation for Deviation from Prior Methods

Our method overall follows similar methods for testing the temperature illusion in traditional non-MR environments. However, we differ by removing the forced-choice method (i.e., "which is warmer?") by not asking participants to decide if an object is hotter or colder than another [12, 13], but enabling them to say it is similar, or the same (by guessing a specific temperature). As we wanted to measure an illusion, we believed it was important for the participant to have the option to see through the illusion, and felt it is perfectly reasonable (or even expected, if the illusion fails) for participants to think the temperature has not changed. Other applied [6] and theoretical research [5, 14] has suggested that forcing choices can cause noise or biases, because there is no means for people to express similarity. Further, our approach of collecting temperature based estimates allows variance to naturally be higher. Thus, if there was a true effect to detect, our approach sets a higher bar for its detection. For these reasons, we decided against forced choice in our experiment design and instead opted for temperature estimation.

### 3.3 Exit Questionnaire

After the experiment, we asked participants to rank 4 colors (Red, Blue, White, and Black) from "coolest" to "warmest" on a 5-point Likert-like scale based on their pre-experiment perception of a color's temperature. We next asked participants to rank the five visual effects they used, based on their experiences during the experiment. We also asked participants to respond to two Likert-scale statements on a 7-point scale: "The fire effect was realistic" and "The ice effect was realistic", with 1=strongly disagree, 4=neutral, 7=strongly agree.

### 3.4 Participants and Data Analysis

We recruited 30 participants (16 self-identified male, 14 female) with an age range of 18 to 45 years (mean = 23.4, sd = 6.163). One participant reported having a slight unspecified difference in sensory ability; however, a cursory visual inspection indicated their results did not differ meaningfully from the other participants'. No participants reported a color-vision deficiency.

A 5x3 (Visual Effect by block) repeated-measures design was used, with block number being treated as a measure of time. Visual effects consisted of the three colors (white, blue and red) and two dynamic-graphic effects (fire and ice). Perceived temperature estimate was the main dependent and was analyzed using RM-ANOVA, the Huynh-Feldt method for adjusting degrees of freedom was used when the assumption of sphericity was violated. Conover's post-hoc tests used Holms' corrections. Perceived temperature rankings were analyzed using Friedman's ANOVA.

## 4 RESULTS

There was a main effect of Visual Effect ($F_{3.35,97.17}$ = 3.16, $p<.05$) on estimated temperature, see Figure 3. Post-hoc tests showed that across all blocks, participants estimated Ice as colder than Blue ($p<.05$). No other differences were significant.

There was an effect of blocks on estimated temperature ($F_{1.93,55.47}$ = 8.05, $p<.005$), see Figure 4. Temperature estimates were significantly lower in block 1 than in blocks 2 and 3.

There was an interaction effect between Visual Effect and block ($F_{6.39,185.19}$ = 2.4, $p<.05$). For simplicity, we present only the differences between visual effects within each block. Within block 1, Fire was estimated as significantly warmer than Ice ($p<.05$) and Ice was significantly colder than Blue ($p<.05$).

### 4.1 Questionnaire Results

For the post-experiment ratings on effect temperature and realism, results showed people ranked colors differently based on their pre-existing association of the colors with temperature ($\chi^2(3)$=47.09, $p<.001$). Post-hoc tests showed that people consistently ranked red

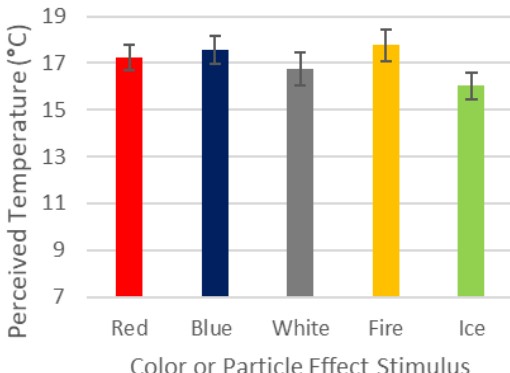

Figure 3: Mean perceived temperatures in Celsius (±SE) across all experimental blocks.

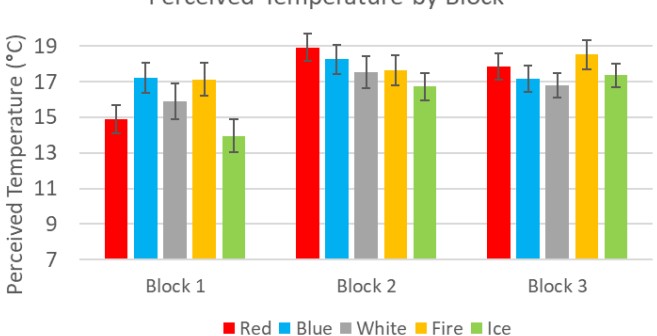

Figure 4: Mean perceived temperature (in degrees Celsius; ±SEM) grouped by trial block, demonstrating the change in temperature perceptions throughout the experiment.

warmer than white ($p<.001$), blue ($p<.001$), and black ($p<.01$). Blue was ranked significantly colder than black ($p<.001$). Survey results also showed that participants perceived temperature (from ranking experimental objects) significantly different ($\chi^2(4)$=19.75, $p<.001$). Post hoc test showed Blue was ranked warmer than Ice ($p<.05$), and Fire warmer than Ice ($p<.001$).

Overall, participants rated the effects only slightly above neutral for realism. The mean agreement that the fire effect was realistic was 4.37 (sd=1.47), and for ice was 4.533 (sd=1.36).

## 5 DISCUSSION

Overall, blue was perceived, on average, as warmer than red, replicating previous reports of the color-temperature illusion. Though we did not find a statistical difference between the red and blue stimuli, the first block shows blue was perceived as roughly 2°C warmer than red while the second two blocks actually show red as being perceived slightly warmer. We attribute the non-statistical results in block 1 for the color stimuli to our task, which asks participants to estimate the temperature rather than use a force ranking task (as in previous work [12, 13, 31]). This suggests that the effect size of the color-temperature smaller when participants are allowed to rate the temperature as similar instead of having a forced to rate it hotter or colder. We also observed the effect reducing over time (in blocks 2 and 3), suggesting that the illusion could be short-lived and had less of an effect as time progressed.

Before the experiment we hypothesized that our dynamic graphic effects would follow the color-temperature illusion (with expecta-

tions leading to inverted perception). Surprisingly, however, our results were the opposite of our expectations. Overall, ice was perceived as colder than blue, and fire was perceived as warmer than ice (in block 1), but fire had the highest mean temperature, and we found it was perceived warmer than ice, which had the lowest mean temperature. This result brings the theorized mechanism underlying the color-temperature illusion into question. Given that the dynamic-graphics illusion seemed to have a stronger visual effect than color, the previously theorized mechanism based on expectations does not seem to hold – if it had, fire should have been perceived as the coldest and ice the warmest. One potential explanation could be the dominant color of the ball itself being white (for ice) and black (the fire ball used a black ball under the fire effect). While the color of the ball could have dominated the particle effect, this is unlikely as the temperature differences of the dynamic-graphic-temperature illusion are still larger than the color-temperature illusion. Further, we selected these colors based on early piloting suggesting the base colors used in the effects provided the most realistic looking. Choosing another color may make the effect look unnatural (less fire-like and ice-like) and reduce the effect. For this reason, we believe our use of dynamic graphic effects has a distinct mechanism to that of color, which is why we believe this dynamic-graphics-temperature illusion is not the same as the color-temperature illusion.

Our results show that more research is needed to uncover the mechanism behind the color-temperature and the dynamic-graphics-temperature illusion. However, our findings also suggest that previously reported illusionary techniques that use scents to lead to temperature illusions might need to consider the important role that dynamic-graphic effects have in creating the experience of potential warm and cold sensations [3].

Results from the first block shows a wider temperature range and variance, but overall lower temperatures in comparison to the latter blocks. No differences except ice and blue were found across all trials, but our interaction and post-hoc tests suggests the illusions weaken in later trials. This is possibly due to participants becoming familiar with the visual effects and being able to could focus more on the thermal feeling, which was explicitly reported by some participants. This suggests that participants eventually became more aware of the true temperatures, weakening the illusions.

We observed evidence of potential sequence effects. Of interest, we saw all temperature estimates begin lower than the real temperature. As block 2 and 3 appear similar when compared to block 1, it is possible that participants experienced a kind of learning effect. Of interest is that all temperatures remained under the actual temperature, and there was still variance between effects. Though no statistical certainty was achieved, this result suggests illusion research that wishes to be applied repeatedly in the real world should specifically test multiple exposures and that more data may be needed to detect smaller effects at future blocks.

Our experiment had sequential and similar stimuli repeated in a short period of time. In a real application, varied effects and illusions may be applied and mixed together, which may affect how the illusion is perceived. Of note is that our stimulus pads were all at room temperature, and how these illusions would affect other temperatures is still unknown. It could be possible that an amplification effect could happen – that a fire effect could amplify the perception of a small actual increase in temperature, which would enable relatively modest cooling and heating requirements in hardware to have larger perceptual effects. Our data did not test this theory, yet our evidence for the this illusion at a standard temperature, along with work suggesting haptics can have a large effect on perception [8, 15] suggests that testing the potential for the illusion to amplify real changes could be important future work.

It is still unclear whether temperature illusions can have a meaningful and measurable effect when put into practice, e.g., in a video game. Indeed, while we found evidence of both the color-temperature illusion and our new dynamic-graphics illusions, many tests failed to find statistical significance, suggesting effects could be small or easily overpowered by other variables. However, given that the dynamic-graphics-temperature illusion provides an intuitive and potentially stronger relationship between the stimulus and the sense of temperature, it is a promising direction and may even be in use in many virtual reality applications, intentional or not. Until now, however, there was little evidence that people may actually be experiencing an illusion when approaching fire or touching ice in mixed reality – future work should study the dynamic-graphics-temperature illusion in both virtual and augmented reality, seek to uncover its underlying mechanism and explore further varieties of particles and textures and investigate if they also produce sensory illusions.

## 6 CONCLUSION

In this work, we sought to explore the application of a well-known illusion, the color-temperature illusion, in augmented reality. Through testing the theorized underlying mechanism of the color-temperature illusion, we created another illusion using dynamic graphics (including particle and texture effects), the dynamic-graphics-temperature illusion. Our results suggest our new temperature illusion works using a different sensory mechanism, leads to an illusion that is stronger in mixed reality than we observed with the color-temperature illusion, and is readily applicable to virtual and augmented reality applications. Our results encourage future research in the design of sensory illusions in mixed and virtual reality, and calls into question some theorized perceptual methods by which these illusions work.

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
