# OpenReview forum: "Temperature Illusions in Mixed Reality using Color and Dynamic Graphics"
_graphicsinterface.org/Graphics_Interface/2023/Conference — GI 2023_

### Official Review · Reviewer_XCTS · 2023-01-13
**interesting problem, potential for followup work**

**Rating:** 7
**Confidence:** 3

**Review:**

This paper presents an investigation into the color-temperature illusion in an augmented reality context. The experiment found weak support for the previously-reported illusion, where warm-colored objects are perceived as cooler than they are, while cool-colored objects are perceived as warmer. Further, and new to this work, the experiment showed a short-lived effect in which the addition of a dynamic texture or particle effect affected perception of temperature.

The topic of illusions in AR is an interesting one and the paper presents a reasonably sound experiment. I recommend acceptance, as I think this has worthwhile findings and can provoke further research. The weak aspect of the paper is the weak and fleeting effect, meaning that this illusion may not be useful in practice. Also, I have some misgivings about the deception and wonder to what extent the findings were influenced by using only room-temperature test objects. (These thoughts are not very concrete and I do not want my hesitation to block the paper's publication.) For me, the identification and preliminary investigation of a new illusion is enough to justify publication, with a minor risk that the effect will not replicate. Many possible comparisons were considered and only a couple were found to be significant. This is sometimes a sign of a nonexistent effect.

Some discussion of the intended uses of illusions in AR/VR would be worthwhile. This discussion could inform further study design.

It might have been worthwhile to include a neutral effect that does not obviously correspond to a temperature difference, in the same way that white objects are perceived as neutral temperature. That would help to untangle the effect of the illusion from the presentation of the visual effect. This might be a direction for future work.

I wonder whether the participants realized that all objects were the same temperature. Some may have suspected that they were, during the first block, and then had that hypothesis semi-confirmed in the subsequent calibration stage. Might it have been a good idea to mix in hot and cold objects into the sequence, even if the temperature estimates of such objects were not intended to be analyzed? One question is whether the dynamic-graphics effect would amplify or reduce the perception of an actual difference from room temperature. It seems as if this illusion would be most useful in practice if it amplified a real difference, so this question is highly relevant.

Were the temperature estimates normally distributed? I cannot tell from the reported analysis. A skewness calculation might be helpful. Space permitting, the full distribution of estimates (not only mean/stdev) could be reported; this would be redundant in the case of a normal distribution, but I would not expect the distribution to be normal.

Minor points:

"An exact definition of illusions is somewhat contentious." This, and the subsequent musing that all perception might be illusion, can be cut. The pragmatic "illusion is perception that does not match reality" is sufficient.

3.2: "the supposed temperature range". At this point the reader has not been informed about the deception, and the word choice is puzzling. Can you reorganize to disclose the deception earlier? It is a key point, and although difficult to miss given the emphasis in the text, could be presented up front so as to allow the reader to interpret the rest of the procedure accordingly.

You went to some trouble to ensure accurate temperatures on the hot and cold blocks, even using an infrared thermometer to ensure that the temperatures were as intended. Why is all this necessary, given the deception of using only room-temperature objects?

The paper mentions at least twice (end of 3.3 and 3.4) that the participants were asked whether the dynamic effects were realistic. (Clearly they are not; no one would mistake the fire effect for real flames.) The results are presented very briefly. This ought to be cut. The effectiveness of the graphic effects does not depend on their realism, as long as the subjects recognize that the effect depicts flames, or depicts some kind of frozen aura. (I am not sure what "realistic" quite means for the icy effect, since it is not clear exactly what natural phenomenon is being referenced.) There does not seem to be any purpose behind the question in any case, since there is no discussion of the implications of the responses; and again, the effects are highly unrealistic and to the extent subjects were unwilling to say so, social desirability bias is responsible.

Equivocating between "significant" (implying "important", "substantial") and "statistically significant" (unlikely to have arisen by chance) is a widespread but undesirable practice. Please avoid phrases like "significantly warmer" to describe statistical significance.

The discussion of "blocks" somewhat obscures that there was only one trial for each condition within a block. This fact is disclosed, but not emphasized. Perhaps some different language could be used or the size of a block reiterated.

Run a spell checker. There are some typos that a spell checker would have caught.

---

### Official Review · Reviewer_7nvY · 2023-01-17
**The paper conducts an experiment on the illusion of temperature perception within a VR environment. The experiment is clear and reproducible and shows novel insights.**

**Rating:** 7
**Confidence:** 3

**Review:**

The paper compares an icy ball to a fiery ball to differently colored balls in a VR setting and measures the temperature perception of the participants. The experiment is clearly described and seems reproducible. It demonstrates significant effects based on the visual depiction, under three different "real" temperatures of the balls.

The paper is well-written and easy to follow. However, there are some minor aspects that could be improved:

- it would be easier to understand if you would replace "Block 1" etc. with the actual condition (7, 21, 33, degrees)

- it is not clear to me why Block 2 shows the highest temperatures (compared to Block 1 and 3) -- was this the condition where the temperature of the ball was 33 degrees?

some typos:

- "the temperature OF objects"
- "and have be studied in experiments where forced choice and relative temperature judgments are made" --> grammar?
- "temperature a illusion" grammar?
- "contains thew same structure" typo

---

### Official Review · Reviewer_P3HJ · 2023-01-20
**Interesting results, but the methods and the discussion need justification and development before publication**

**Rating:** 4
**Confidence:** 4

**Review:**

This paper presents a study looking at colour-temperature illusion in VR, The study investigated 5 conditions: three static colors (blue, red, white) and two dynamic colors (ice, fire), where participants estimated the temperature after an anchoring of 7ºC, 21ºC, and 33ºC. The experiment was conducted in three blocks. The results find that, for the first block, static color illusions match prior work (with the counter-intuitive outcome that blue sensations feeling warm, while red illusions feel cold), white was neutral, but ice and fire sensations show the opposite effect: fire is warm, ice is cool. In blocks 2 and 3, we see a learning effect where temperature estimations change over the course of the experiment.

I unfortunately cannot argue for acceptance with the paper as written, as I believe it needs iteration before publication. I believe the finding is very interesting, and I hope to see this work published some day. However, I have concerns about the validity of the methods, the degree of discussion and analysis into the findings, and from those two concerns, the trustworthiness and size of the contribution. I also think that the paper needs more details before it can be published.

Methods: The methods used do not draw upon any sort of psychophysics techniques common in the area. I do not believe that this work needs psychophysics, but the methods used seem ad-hoc and throw out forced-choice methods without much rationale. I believe this paper needs to justify its methods, showing at least some awareness of methods used by prior work for perceptual studies to make a case that its approach is valid. The authors do cite [5] to argue against forced-choice methods, but I am not convinced that is sufficient rationale: [5] is a study involving forced choice vs a non-choice option in purchasing decisions. I need more explanation about how this impacts perceptual studies, which (I believe) have a different paradigm. See [a] below for more details. The authors might want to pay special attention to the method of "magnitude estimation" which is closest to their method.

I am also concerned about a possible confound with the white ball used for both the neutral condition and the ice condition, and the fire condition using a black ball. The readers need additional justification for these decisions - why not use blue and red for ice and fire to really isolate the dynamic vs. static condition? The construct validity is weak here. If it were stronger, planned contrasts could help us analyze across the color and dynamic conditions, possibly.

Analysis and Discussion: Even if the methods are valid, I think that the outcomes of this work need more attention. There is a short analysis of the overall effects, and a plot of the learning effects. In block 1 we see a very clear outcome, but in blocks 2 and 3 the effect gets washed out. I think the learning effect is an important consideration, but it might be worth analyzing just the first condition. With enough exposure, participants might learn that all the temperatures at the same, especially because given uncertainty, there's evidence that people trust the haptic sense over visual [b]. I think the implications here are critical for the general application of these illusions in VR, as it might invalidate the entire technique. More analysis should be done, with more discussion about the implications.

Writing: There are several typos, and the the paper would benefit from more careful explanation of the different conditions, including rationale and images of each. The terminology is inconsistent; I had to search to figure out that "black" meant "fire", as the fire condition used a black ball.

[a] Jones LA, Tan HZ. Application of psychophysical techniques to haptic research. IEEE transactions on haptics. 2012 Dec 13;6(3):268-84.
[b] Fairhurst MT, Travers E, Hayward V, Deroy O. Confidence is higher in touch than in vision in cases of perceptual ambiguity. Scientific Reports. 2018 Oct 23;8(1):1-9.

---

### Meta-Review · Area_Chair_3ZWb · 2023-01-20

**Recommendation:** 6
**Confidence:** 3

**Metareview:**

The paper received three reviews with scores of 4, 7, and 7 with reviewer confidence of 4, 3, and 3. Especially the reviewer with a more critical review cites higher confidence. All reviewers, however, have suggestions and concerns about the user study and its analysis. Therefore, in case of acceptance, we suggest the improve these aspects of the paper:
* a discussion of the appropriateness of the method (why not forced choice)
* a better justification of the white/black ball basis condition
* a more thorough analysis
* a more detailed report of the results

All reviewers agree that the results (and the work) are novel and exciting to a broader research community.